

# Alteration of the oral and gut microbiota in patients with Kawasaki disease

Qinghuang Zeng[1,2], Renhe Zeng[1,2] and Jianbin Ye[3,4]

[1] School of Clinical Medicine, Fujian Medical University, Fuzhou, China
[2] Affiliated Hospital of Putian University, Putian, Fujian, China
[3] School of Basic Medicine Science, Putian University, Putian, China
[4] School of Pharmarcy, Fujian Medical University, Fuzhou, China

## ABSTRACT

**Background**. Kawasaki disease (KD) is a multi-systemic vasculitis that primarily affects children and has an unknown cause. Although an increasing number of studies linking the gut microbiota with KD, the unchallengeable etiology of KD is not available.

**Methods**. Here, we obtained fecal and oral samples from KD patients and healthy controls, and then we use high-throughput sequencing to examine the diversity and composition of microbiota.

**Results**. Results showed that both in the gut and oral microbiota, the diversity of KD patients was significantly lower than that of the healthy controls. In the gut microbiota, a higher abundance of *Enterococcus* (40.12% *vs* less than 0.1%), *Bifidobacterium* (20.71% *vs* 3.06%), *Escherichia-Shigella* (17.56% *vs* 0.61%), *Streptococcus* (5.97% *vs* 0.11%) and *Blautia* (4.69% *vs* 0.1%) was observed in the KD patients, and enrichment of *Enterococcus* in the patients was observed. In terms of oral microbiota, the prevalence of *Streptococcus* (21.99% *vs* 0.1%), *Rothia* (3.02% *vs* 0.1%), and *Escherichia-Shigella* (0.68% *vs* 0.0%) were significantly higher in the KD patients, with the enrichment of *Streptococcus* and *Escherichia-Shigella*. Additionally, significant differences in microbial community function between KD patients and healthy controls in the fecal samples were also observed, which will affect the colonization and reproduction of gut microbiota.

**Conclusions**. These results suggested that the dysbiosis of gut and oral microbiota are both related to KD pathogenesis, of which, the prevalence of *Enterococcus* in the gut and higher abundance of *Streptococcus* and *Escherichia-Shigella* in the oral cavity will be a potential biomarker of the KD. Overall, this study not only confirms that the disturbance of gut microbiota is a causative trigger of KD but also provides new insight into the oral microbiota involved in KD pathogenesis.

Corresponding authors
Renhe Zeng, 360153392@qq.com
Jianbin Ye, happye1986@163.com

## INTRODUCTION

Kawasaki disease (KD), which was first reported by Dr. Tomisaku Kawasaki, is an acute febrile illness of early childhood. The age-specific incidence rate of KD is highest in children under 1 year old (*Uehara & Belay, 2012*). The KD has now become one of the leading causes of acquired heart disease in children in developed countries because its' pathology is self-limited vasculitis that primarily involves the coronary arteries. The prognosis of KD

depends on the extent of cardiac involvement, and up to 20%–25% of patients will develop coronary aneurysms (*McCrindle et al., 2017*). It has been reported in children of all races although it was first reported in Japan.

However, the etiology of KD remains obscure. The clinical and epidemiological features suggest an infectious origin or trigger. The clinical features of KD are a self-limited illness including fever, rash, mucositis, *etc.*, which appear like peculiar infectious diseases, such as streptococcal infections, and atypical measles (*Burgner & Harnden, 2005*; *Esposito, Polinori & Rigante, 2019*). In addition, the seasonality of cases (*Nakamura et al., 2008*), the age distribution of KD (*Uehara & Belay, 2012*), the occurrence of community outbreaks, and the high incidence rate in siblings of KD (*Kinumaki et al., 2015*) implied that this disease is transmissible in children. Based on these, the reasonable theory behind the disease is that KD may be triggered by one or more infectious agents (*Newburger et al., 2004*). Nevertheless, no exact causative agent has been identified so far, although researchers have tried their best to isolate the bacteria, viruses, and fungi with conventional inoculation.

Meanwhile, some genome-wide association studies (GWAS) revealed susceptibility loci involved in immune disorders and cardiovascular status (*Burgner et al., 2009*; *Onouchi et al., 2012*). Many studies have demonstrated that the immune system plays an important role in KD patients. For example, the levels of chemokines and cytokines, such as interleukin (IL)-1, IL2, Il-6, IL-17, IL-23, and tumor necrosis factor-$\alpha$ are reported to be elevated in the acute phase of patients (*Greco et al., 2015*; *Jia et al., 2010*). The elevations of neutrophils (lipopolysaccharide binding), plasma proteins, and antibody reactivity against mycobacterial heat-shock protein (HSP60) were observed in the convalescent sera in several previous reports (*Kinumaki et al., 2015*; *Takeshita et al., 2002b*). These factors are mostly related to the secondary infections of some pathogens, including bacteria (*Streptococcus pyogenes*, *Klebsiella pneumoniae*, *Staphylococcus aureus*) (*Principi, Rigante & Esposito, 2013*) and viruses (Epstein-Barr virus, parvovirus B19, rotavirus, dengue virus, and influenza virus) (*Joshi et al., 2011*; *Principi, Rigante & Esposito, 2013*). It is worth noting that the relationship between KD and coronavirus disease 2019 (COVID-19) was also recently documented due to the pandemic, and the hyperinflammation induced by COVID-19 may act as a primer for KD development in some patients (*Moreira, 2020*; *Verdoni et al., 2020*; *Viner & Whittaker, 2020*). These extensive studies promote the identification of the pathogenesis and pathophysiology of KD and further suggest that infectious agents induce the onset of the disease.

The dysregulation of the gut microbiota may lead to autoimmune diseases through aberrant immune system development. For adults, the intestinal microbiota is considered to be inter-individually variable and intra-individually stable, which plays a crucial role in protecting the mucosal immune system. However, the gut microbiome of infants and children is more easily disturbed by diet, formula feeding, and other life events, including infection and antibiotic treatment (*Kinumaki et al., 2015*; *Morotomi et al., 2011*). Thus, more and more researchers are focused on the link between the intestinal microbiota and patients with KD. The KD children often have gastrointestinal symptoms and complications (*Eladawy et al., 2013*). Based on the cultured methods, some previous studies have identified possible causative microbial agents of KD, such as Gram-negative bacteria capable of

producing HSP60 and Gram-positive cocci that induce the V $\beta$2T cell expansion. Another study suggested that the lack of *Lactobacilli* (*Takeshita et al., 2002a*) during the acute phase of KD patients could be a possible reason. Along with the development of metagenomic analyses, several studies sought to find more relationships between uncultured intestinal microbiota and KD. In 2015, *Kinumaki et al. (2015)* first analyzed the fecal microbiome of 28 KD patients by using metagenomics and revealed that *Streptococcus* spp. were more abundant in the acute phase and suggested KD-related streptococci might be involved in the pathogenesis of KD. Recently, Kaneko et al. raised a new viewpoint that dysbiosis is the pathogenesis of KD, and more information on the gut microbiota in the feces of antibiotic-naive KD patients is needed (*Kaneko et al., 2020*).

Nevertheless, few studies have focused on the oral microbiota of KD patients. More than 700 kinds of microorganisms were colonized in the human oral cavity, and large studies were carried out to elucidate the relationship between oral microbes and human diseases, including diabetes, preterm birth, and cardiovascular diseases, etc. (*Gomez et al., 2020*; *Li et al., 2021*; *Matsha et al., 2020*; *Xian et al., 2018*). The disturbance of oral microbes in systemic diseases is repeatable, which suggests that oral microbes can reflect the status of disease and health in real-time and are of great value in disease risk early warning and curative effect prediction (*Peng et al., 2022*). Besides, the oral administration of *Porphyromonas gingivalis* (a representative periodontopathic bacterium) was demonstrated to alter the gut microbiota composition (dysbiosis) by reducing gut barrier function and modulating the gut immune system (*Kato et al., 2018*). Increasing evidence suggests that oral microbes may alter the gut microbiome by invading the intestine, causing imbalances in the microecology and affecting the digestive system (*Gao et al., 2018*). Thus, obtaining the oral microbes of KD patients is of great value to elucidate the etiology of KD.

This study aims to compare differences in oral and gut microbes between KD patients and healthy controls. A comparative metagenomic approach was used, and the information could be a complement to the understanding of the microbiome and infections in the pathogenesis of KD.

## MATERIALS & METHODS

### Samples collection

Ten samples of children (five from the KD patients, and other five from the healthy) without antibiotic treatment were collected from the Affiliated Hospital of Putian University. All of the KD patients enrolled in this study were collected in the acute phase before admission, both fecal and oral samples were collected on the first day they arrived hospital. KD diagnosis meets the criteria established by the KD Research Committee of American Heart Association (*McCrindle et al., 2017*). The controls were collected from the healthy children and were not treated with any medicine for 1 month before sample collection. Both fecal and oral samples were collected. Finally, one fecal sample of KD patients was excluded due to poor sample quality after DNA amplification. Thus, only nine fecal samples (five from the patients and four from the healthy controls) and 10 oral samples (five from the patients and five from the healthy controls) were subjected to the following sequencing
and analysis. Both the KD patients and healthy controls were collected with fecal and oral samples. The basic information of all samples were listed in Table S1. Samples were stored in the 5 ml tubes and immediately frozen at −80 °C until use. This study was approved by the ethics committee of the Affiliated Hospital of Putian University (No: 202183), and the informed consent was signed by parents.

## DNA extraction and amplification

Total DNA extraction was performed using the E.Z.N.A.® soil DNA Kit (Omega Biotek, Norcross, GA, USA) according to the manufacturer's instructions. DNA quality and concentration were detected using the NanoDrop one UV–vis spectrophotometer (Thermo Fisher Scientific, Waltham, MA, USA). By using the primers 338F (5′-ACTCCTACGGGAGGCAGCAG-3′) and 806R (5′-GGACTACHVGGGTWTCTAAT-3′), the V3-V4 region of the bacterial 16S rRNA gene, which has about 465 bases, was amplified. The PCR settings were as followings: initial denaturation at 95 °C (3 min), then 27 cycles of denaturing at 95 °C (30 s), annealing at 55 °C (30 s), and extension at 72 °C (45 s), followed by a single extension at 72 °C (10 min), and finish at 4 °C. Each PCR had the following ingredients: 5× *TransStart* FastPfu buffer 4 µL, 2.5 mM dNTPs 2 µL, 0.8 µL each of the forward primer and reverse primers (5 µM each), *TransStart* FastPfu DNA Polymerase 0.4 µL, template DNA 10 ng, and finally ddH$_2$O up to 20 µL. A triplicate of each PCR reaction was run.

## Illumina Miseq sequencing

PCR products were purified by using the AxyPrep DNA Gel Extraction Kit (Axygen Biosciences, Union City, CA, USA) and quantified using Quantus™ Fluorometer (Promega, USA) both following the instructions of the manufacturer. Purified amplicons were pooled in equimolar amounts and paired-end sequencing were performed on an Illumina MiSeq platform (Illumina, San Diego, USA) according to the standard protocols by a commercial company (Majorbio Bio-Pharm Technology Co. Ltd., Shanghai, China) using the MiSeq Reagent Kit v3. The raw reads were deposited into the NCBI Sequence Read Archive (SRA) database (accession number: PRJNA928221).

## Processing of sequencing data

Fastp version 0.20.0 (*Chen et al., 2018*) was used to quality-filter the raw data of paired-end sequences after it had been first assembled using FLASH version 1.2.11 (*Magoc & Salzberg, 2011*). Sequencing reads with exact matches to barcodes was recognized as authentic sequences and assigned to the appropriate samples after the potential chimera sequences were filtered. Using UPARSE version 7.1 (*Edgar, 2013*), operational taxonomic units (OTUs) with a 97% similarity criterion were grouped, and chimeric sequences were found and eliminated. Using a confidence threshold of 80%, RDP Classifier version 2.2 (*Wang et al., 2007*) was used to assess the taxonomy of each OTU representative sequence against the 16S rRNA database (eg. Silva v138).

## Statistical analysis

The alpha-diversity (including Chao index, Shannon index, and Simpson index) of bacteria was performed by using Mothur version 1.30.1 (*Schloss et al., 2009*). The differences

between groups were displayed by the Partial least square discriminant analysis (PLS–DA) and nonmetric multidimensional scaling (NMDS) based on Bray–Curtis dissimilarity. The differences between the case and healthy control groups were analyzed by the Analysis of similarity (ANOSIM). Statistical significance was defined as a $P$-value of less than 0.05.

## RESULTS

### Differences in microbial diversity among KD patients and healthy controls

In total, nine fecal specimens (five KD patients and four healthy control) and 10 oral specimens (five KD patients and five healthy control) were sequenced successfully. After quality control processes of sequences, 836,786 sequences were obtained from all samples, with an average of 44,041 sequences per sample (Table S2). A total of 495 OTUs were obtained from all samples, including 262 OTUs and 314 OTUs from fecal samples and oral samples, respectively. Compared to the KD patients, higher OTUs were obtained in the healthy control group both in oral (258 in control $vs$ 238 in KD patients) and fecal samples (219 in control $vs$ 175 in KD patients) (Fig. S1). The rarefaction curve showed that the sequencing depth was sufficient, as the curve reached a plateau in all samples (Fig. S2), indicating that most microbiota species were captured. Alpha diversity indices were calculated to evaluate the microbiota's richness and diversity. Although slightly higher microbiota richness was observed in the healthy controls, no significant differences in richness were detected between the two groups ($P = 0.066$, >0.05) neither in oral samples nor in fecal samples. This may be attributed to the small number size of samples, as the error bar of the Sobs index was high (Fig. 1B). However, the diversity from KD patients was significantly lower than that from the healthy controls ($P = 0.0199$, <0.05). Similarly, a significantly lower Shannon index of oral bacteria was observed in the KD patients ($P = 0.0216$, <0.05) as well (Fig. 1). These results suggested that KD diseases could induce the loss of biodiversity in both the oral and gut microbiomes.

The difference between the two groups of KD patients and healthy controls was evaluated *via* beta diversity by performing PLS-DA and NMDS based on Bray–Curtis dissimilarity (Fig. 2). Although the number of samples was small, the results showed that the microbiota of KD patients was distinct from that of healthy controls both in the oral and fecal samples ($R = 0.7527$, $P = 0.001$, ANOSIM). The separation trend of the fecal samples was more significant than that of the oral samples. PCA and PCoA also demonstrated the distinction between the four groups (Fig. S3A). Furthermore, the heatmap of sample hierarchical clustering showed that most of the samples from each group could be clustered well based on microbiota at the genus level (Fig. S3B).

### Differences in the composition of bacterial communities in KD patients and healthy controls

The taxonomic composition of bacteria was analyzed from phyla to genera and visualized in the form of a bar diagram (Fig. 3). The five major phyla in all groups were Firmicutes, Bacteroidota, Proteobacteria, Actinobacteria, and Fusobacteriota. However, the distribution of these phyla abundances was different between KD patients and healthy

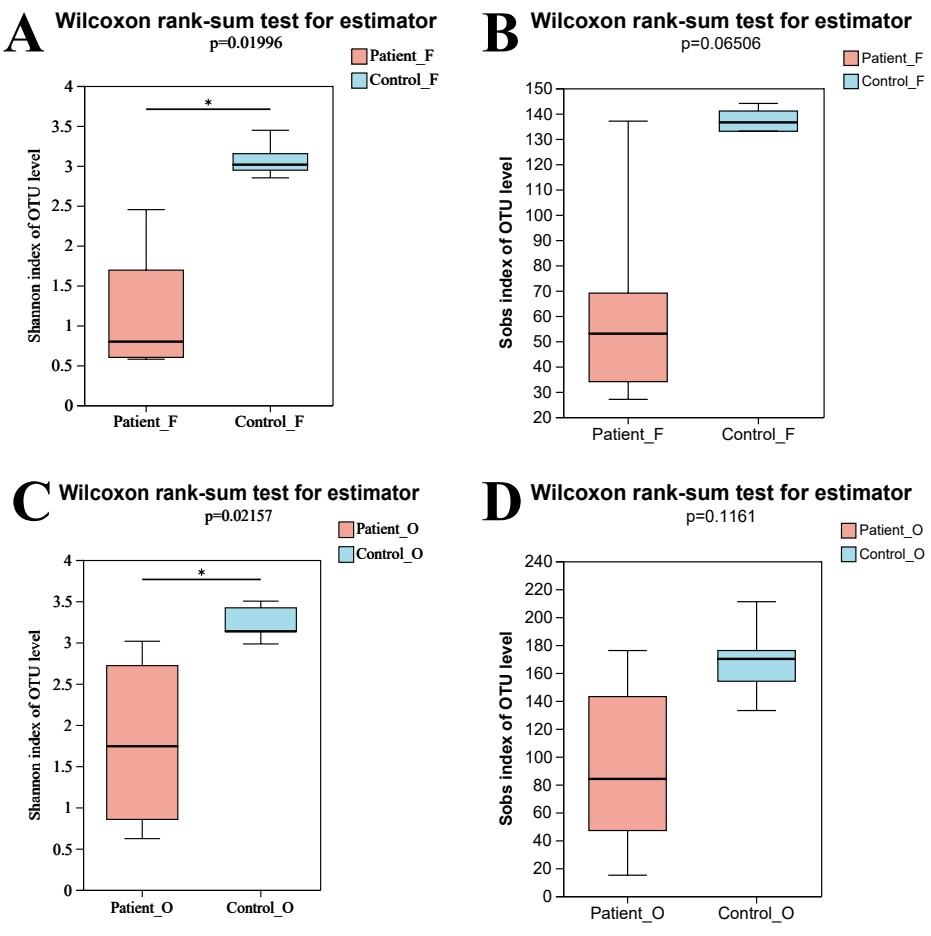

**Figure 1 The alpha diversity indices of oral and gut microbiota.** Shannon index of fecal samples (A). Sobs index of fecal samples (B). Shannon index of oral samples (C). Sobs index of oral samples (D). Patient F and Control F represent fecal samples from KD patients and healthy, respectively; Patient O and Control O represent oral samples from KD patients and healthy, respectively.

controls (Fig. 3A). In fecal samples, *Firmicutes* (59.5%) was first dominant in the patients, followed by *Actinobacteria* (22.0%) and *Proteobacteria* (17.8%). On the contrary, the *Bacteroidota* (47.9%) was the first dominant in the healthy control, with an obvious decrease of *Firmicutes* (44.1%) and a significant decreasing of *Actinobacteria* (4.25%) and *Proteobacteria* (3.4%). This indicated that the decreasing of *Bacteroidota* could be one of the factors related to KD disease. In oral samples, *Firmicutes* was also the dominant phylum in healthy controls (41.4%), but with an obvious increase in the KD patients (71.2%). Compared to healthy controls, the *Proteobacteria* (21.79% of healthy controls *vs* 6.08% of KD patients), *Bacteroidota* (15.71% of healthy controls *vs* 8.17% of KD patients), and *Fusobacteriota* (9.70% of healthy controls *vs* 2.41% of KD patients) were all decreased in the KD patients, except for the slightly increasing of *Actinobacteria* (9.4% of healthy controls *vs* 11.7% of KD patients) (Fig. S4).

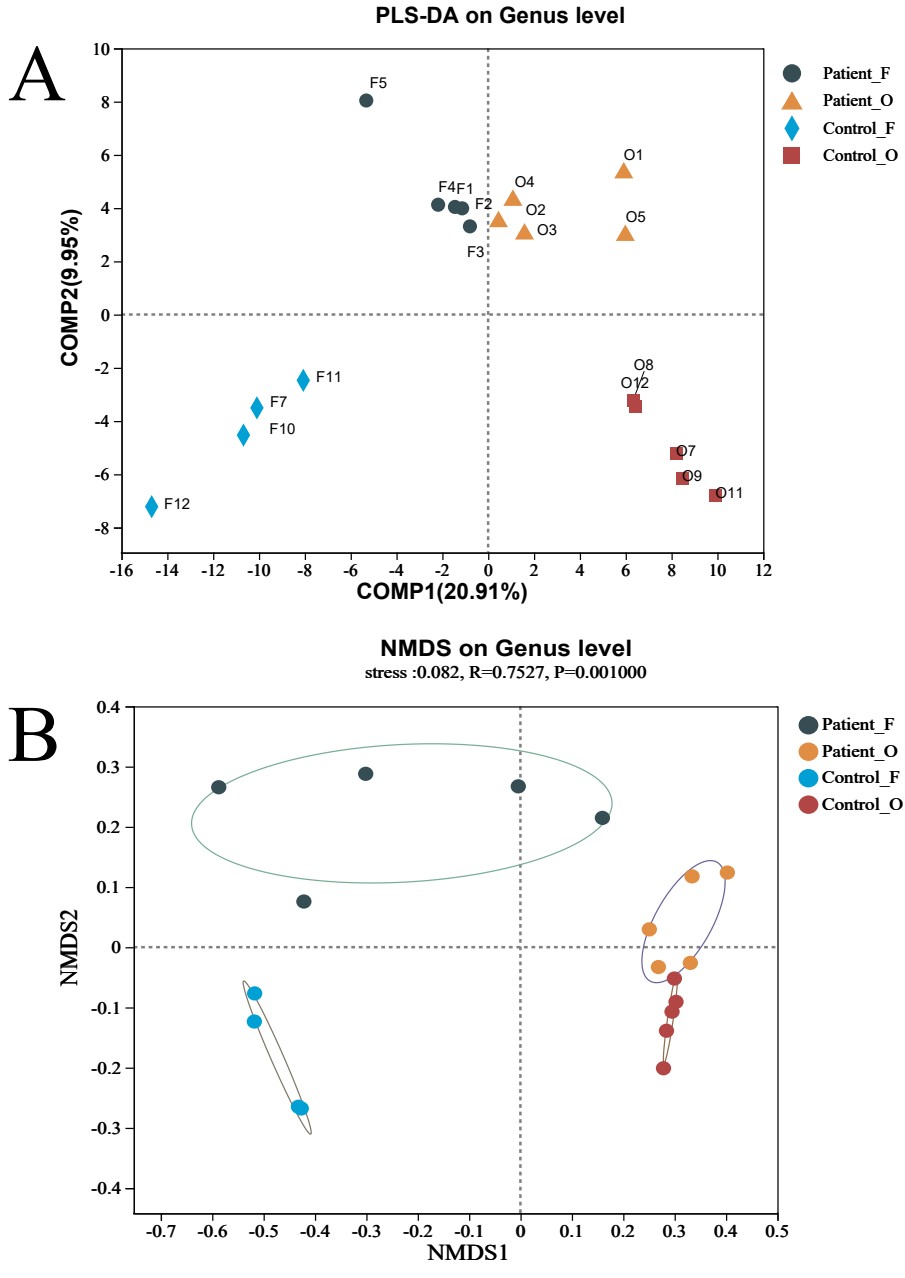

**Figure 2** (A–B) **The beta diversity of all groups.** Patient F and Control F represent fecal samples from KD patients and healthy, respectively; Patient O and Control O represent oral samples from KD patients and healthy, respectively.

At the genus level, large differences were observed between the KD patients and healthy controls (Fig. 3B). In the fecal samples of KD patients, the main genera of the microbiome include *Enterococcus* (40.12%), *Bifidobacterium* (20.71%), *Escherichia-Shigella* (17.56%), *Streptococcus* (5.97%) and *Blautia* (4.69%). However, the dominant genera in the fecal samples of healthy controls included the *Bacteroides* (41.29%), *Faecalibacterium* (7.19%),

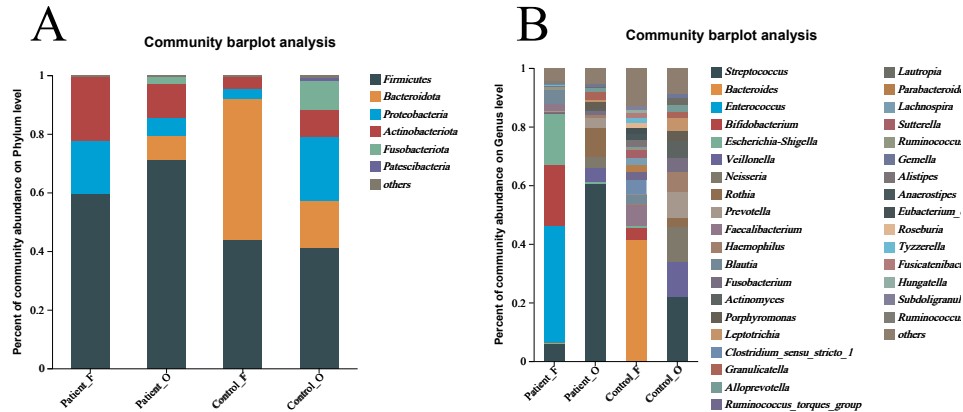

**Figure 3** The relative abundance of bacterial communities at the phylum (A) and genus (B) levels. Patient F and Control F represent fecal samples from KD patients and healthy, respectively; Patient O and Control O represent oral samples from KD patients and healthy, respectively.

*Bifidobacterium* (4.17%), and *Blautia* (3.06%). In comparison, the relative abundance of *Enterococcus* (less than 0.1%), *Escherichia-Shigella* (0.61%), and *Streptococcus* (0.11%) in fecal samples was significantly lower in the healthy controls. Interestingly, *Streptococcus* (60.49%) was the first dominant genus in the oral samples of KD patients, followed by *Rothia* (10.09%), *Veillonella* (4.91%), Neisseria (3.49%), *Prevotella* (3.15%) and *Escherichia-Shigella* (0.68%). Compared to the KD patients, the relative abundance of *Streptococcus* (21.99%) and *Rothia* (3.02%) was significantly lower in the oral samples of healthy controls. Not surprisingly, the other genera were all increased in the healthy controls, including *Veillonella* (11.94%), *Neisseria* (11.99%) and *Prevotella* (8.88%), *Haemophilus* (6.68%), *Actinomyces* (5.57%) and *Fusobacterium* (5.10%) (Fig. S5). These results suggested that the microbiota in oral samples from healthy controls is more diverse.

## Potential bacterial biomarkers of KD patients

The LEfSe analysis at the genus level (LDA score>2.0, $p < 0.05$) was performed to find the potential bacterial biomarkers related to the KD disease. In the fecal samples, only the *Enterococcus* was significantly enriched in KD patients, with the SCFA-producing microbiota including *Bacteroides*, *Faecalibacterium*, *Clostridium*, *Parabacteroides,* and *Prevotella* significantly reduced in the KD patients (Fig. 4A). It is worth noting that the *Streptococcus* and *Escherichia-Shigella* (although the relative abundance of *Escherichia-Shigella* in the KD patients was only 0.68%) were enriched in the oral samples of KD patients, while the *Haemophilus*, *Actinomyces*, *Leptotrichia* and other genera were enriched in the healthy controls (Fig. 4B).

## Functional capability analysis

The functional gene composition of microbiota in oral and fecal samples was analyzed by the PICRUSt2 (T student test) to differentiate the normal biological function between KD patients and healthy controls. In total, 263 KEGG Level 3 modules were obtained

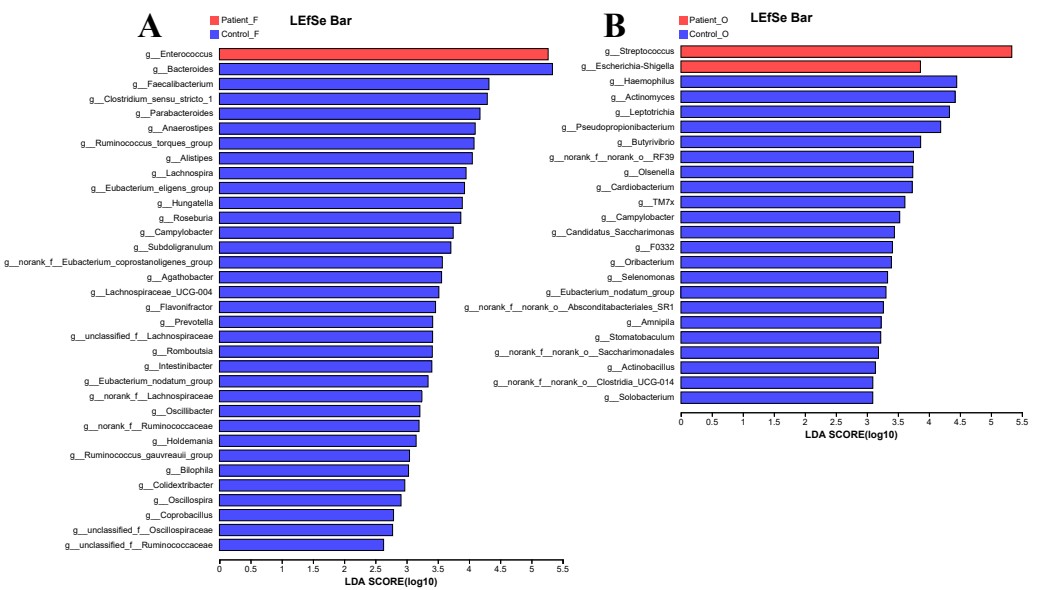

**Figure 4  LEfSe at genus level: fecal samples (A) and oral samples (B).** Patient F and Control F represent fecal samples from KD patients and healthy, respectively; Patient O and Control O represent oral samples from KD patients and healthy, respectively.

for all samples. For KEGG level 1, no significant differences were observed between the KD patients and healthy controls neither in oral nor in fecal samples (Figs. 5A, 5B). However, when looking into the KEGG level 2 (Fig. 5C), Xenobiotics biodegradation and metabolism and Membrane transport were significantly higher in the fecal samples of KD patients ($P < 0.05$), while the immune system, environmental adaptation, transport, and catabolism and glycan biosynthesis and metabolism were much higher in the fecal samples of healthy controls ($P < 0.05$). Although some differences in the KEGG pathway at level 2 were observed in oral samples between KD and healthy controls (Fig. 5D), no significance was detected as the corrected $P$ value was higher than 0.05 ($p > 0.05$).

## DISCUSSION

The microbiota plays a crucial role in human health and physiology, and dysbiosis of microbiota has been demonstrated to be associated with immune-mediated disorders, rheumatologic diseases, infections, and disorders of the nervous system (*Esposito, Polinori & Rigante, 2019*). Indeed, the relationship between the microbiota of humans and the immune system is very close. The microbiota inhabits the intestine, oral cavity, skin, and other surfaces of the human body and has an impact on human health and also on the prevention of disease (*Esposito, Polinori & Rigante, 2019*; *Robinson & Pfeiffer, 2014*). Nonetheless, the relationship between microbiota and Kawasaki syndrome is not fully understood, although a few studies have demonstrated that the alteration of gut microbiota is associated with KD pathogenesis (*Chen et al., 2020*; *Shen et al., 2020*). For example, several infectious agents (including bacteria and viruses) have been demonstrated

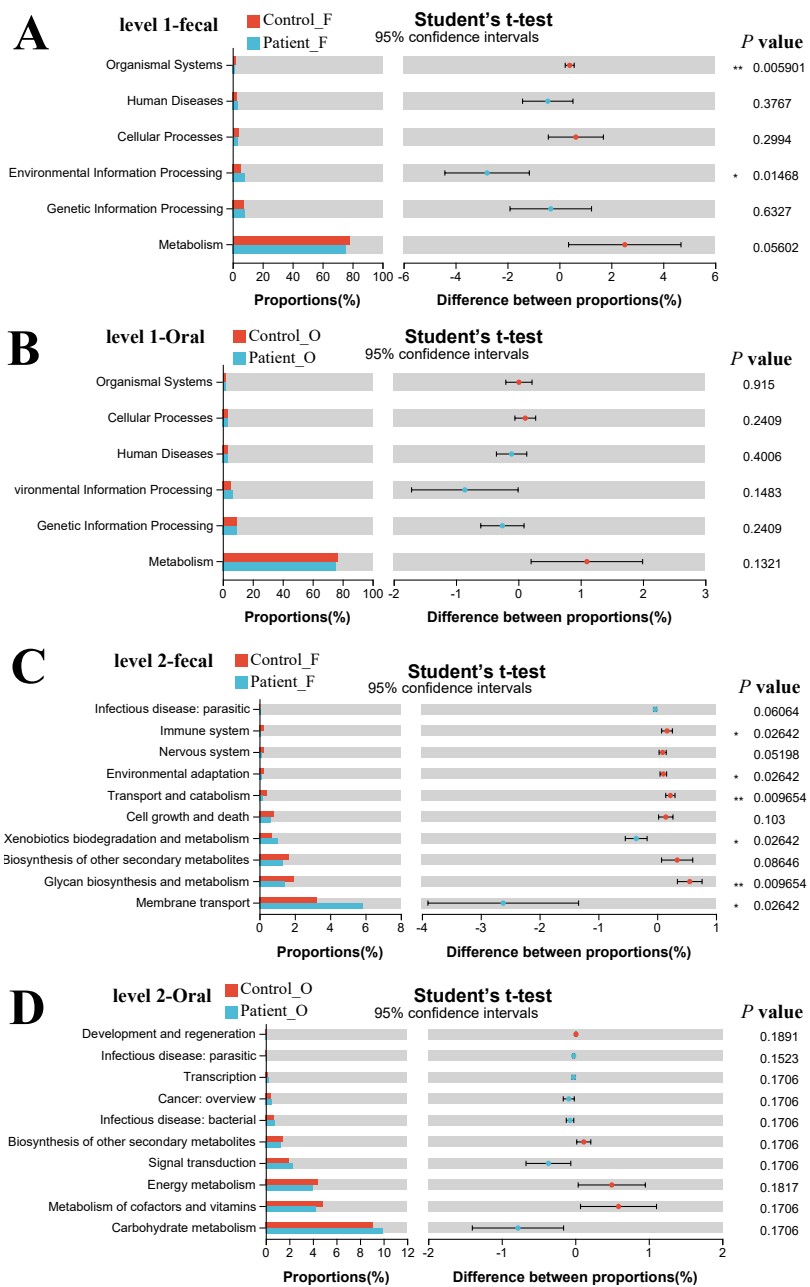

**Figure 5   Functional capability analysis based on the KEGG pathways. Level 1-fecal (A), Level 1-oral (B), Level 2-fecal (C), Level 2-oral (D).** Patient F and Control F represent fecal samples from KD patients and healthy, respectively; Patient O and Control O represent oral samples from KD patients and healthy, respectively.

to be associated with KD pathogenesis (*Joshi et al., 2011*; *Kinumaki et al., 2015*), but these speculations need to be further confirmed by more data.

Increasing evidence suggests that the gut microbiota of healthy individuals is different from that of KD patients. *Kinumaki et al. (2015)* first reported the gut microbiota of KD

children by using metagenomic sequencing technology. The composition of gut microbiota was significantly different between the acute and non-acute phases of KD patients and five *Streptococcus* spp. (*S. pneumonia*, *pseudopneumoniae*, *oralis*, *gordonii*, and *sanguinis*) were markedly increased in the acute phase patients (*Kinumaki et al., 2015*). The authors believed that gut microbiota disorders were closely related to KD pathogenesis. To note, this former study only investigated the gut microbiota of KD patients at different phases (acute and non-acute) without a healthy control group. Although only 10 samples were collected from KD patients and healthy controls due to the small number of volunteers, differences in microbiota between KD patients and healthy controls were still observed in the present study. Other previous studies revealed that KD patients seem to exhibit a disorder of gut microbiota, and the decreasing of microbial diversity was commonly observed in KD patients when compared with the healthy controls (*Khan et al., 2020*; *Shen et al., 2020*). Here, our study also suggested that the diversity of gut microbiota in KD patients is significantly lower than that of healthy controls. Beta diversity showed that the compositions of gut microbiota in the healthy controls were distinct from those in the KD patients as well.

Nevertheless, the relationship between oral microbes and KD has not been reported. Similar to the gut microbiota in our study, a significantly lower biodiversity of oral microbiota was observed in KD patients, and the beta diversity indicated that the microbiota was also significantly different between KD and healthy controls. These results implied that the gut microbiota and the oral microbiota could be involved in KD. Oral microbial infection is demonstrated to affect other parts of the human body, which is relevant to various systemic diseases (*Falcao & Bullon, 2019*). Many studies have been convinced that the changes in oral microbiota are associated with human chronic diseases, including cancer (*Tuominen & Rautava, 2021*), diabetes (*Matsha et al., 2020*), and inflammatory bowel disease (IBD) (*Read, Curtis & Neves, 2021*). Indeed, evidence suggested that changes in oral microbiota could induce the dysbiosis of gut microbiota by reducing gut barrier function and modulating the gut immune system (*Gao et al., 2018*; *Kato et al., 2018*). These effects could finally lead to the occurrence of autoimmune diseases, such as KD. Given that KD is most probably triggered by infectious agents (*Newburger et al., 2004*), the differences in oral microbiota between KD patients and healthy controls should receive more attention. Thus, more oral samples should be collected from KD patients to further elucidate the relationship between oral microbiota and KD. In addition, the lack of information on the level of inflammatory markers in the present study should also be considered in the future.

It is worth noting that some of the human pathogens were significantly increased in KD patients. In the fecal samples, a higher abundance of *Enterococcus*, and *Escherichia-Shigella* was observed in the gut microbiota of KD patients compared with healthy controls. Some *Escherichia-Shigella* was the pathogenesis of shigellosis (the clinical symptoms included watery diarrhea, inflammatory bacillary dysentery with abdominal cramps, and fever) (*Schroeder & Hilbi, 2008*). The *Enterococcus* pathogens usually result in nosocomial infections, including those of the urinary tract, peritoneum, and respiratory tract (*Treitman et al., 2005*). Meanwhile, *Enterococcus* has a high capacity for biofilm formation. The biofilms produced by *Enterococcus* can activate the body to secrete some superantigens and

induce a strong inflammatory response which was considered a possible pathogenesis of KD (*Guiton et al., 2010*; *Kusuda et al., 2014*). Here, we notice that *Enterococcus* was significantly enriched in the gut microbiota in KD patients, which indicated it as a potential biomarker of this disease. Similarly, *Streptococcus* and *Escherichia-Shigella* were enriched in the oral samples of KD patients. *Streptococcus* has been widely identified in KD patients and demonstrated to possess potential superantigenic properties, but most of them have been reported in the gut microbiota (*Khan et al., 2020*; *Kinumaki et al., 2015*; *Nagata et al., 2009*). We noticed that the *Streptococcus* in gut microbiota is also relatively higher in the KD patients compared with the healthy controls (5.97% *vs* 0.11%) in this study. To date, our study is the first to report that *Streptococcus* in the oral microbiota might be involved in the pathogenesis of this disease. This result further supports that the differences in oral microbiota between KD patients and healthy controls could be a cause of alteration of gut microbiota.

Our study also observed significant differences in microbial community function between KD patients and healthy controls in the fecal samples. In KD patients, the immune system, environmental adaptation, transport and catabolism, and glycan biosynthesis and metabolism were weakened, which agreed with former studies suggesting that this metabolism could affect the colonization and reproduction of gut microbiota (*Chen et al., 2020*). Xenobiotic biodegradation and metabolism and membrane transport were enhanced in the KD patients which may be related to the enrichment of *Enterococcus* in the gut microbiota.

## CONCLUSIONS

In this study, we report that the diversity of microbiota in the gut and oral environment will be significantly reduced in KD patients compared with healthy controls. Additionally, the prevalence of *Enterococcus*, *Escherichia-Shigella* in gut microbiota, and *Streptococcus* in the oral cavity could be potentially involved in the KD pathogenesis. Our results confirmed that gut microbiota is related to KD pathogenesis and first provided the new view that oral dysbacteriosis is another etiology of KD. These findings explore our understanding of the KD pathogenesis and new treatments of this disease. However, larger samples are necessary and the relationship between the inflammatory markers and the microbiota should also be considered in future studies.

## ACKNOWLEDGEMENTS

We thank the employees of the Affiliated Hospital of Putian University for their help with sample collections. We also thank to the reviewers and editors for the constructive comments to improve our manuscript.

### Funding

This work was supported by the Science and Techonoly Project of Putian City (2021S3F001, 2022SZ3001ptxy08, 2022SZ3001ptxy09) and the Natural Science Foundation of Fujian Province (2021J05242, 2021J05241, 2022J011167). The funders had no role in study design, data collection and analysis, decision to publish, or preparation of the manuscript.

### Grant Disclosures

The following grant information was disclosed by the authors:
The Science and Techonoly Project of Putian City: 2021S3F001, 2022SZ3001ptxy08, 2022SZ3001ptxy09.
The Natural Science Foundation of Fujian Province: 2021J05242, 2021J05241, 2022J011167.

### Competing Interests

The authors declare there are no competing interests.

### Author Contributions

- Qinghuang Zeng conceived and designed the experiments, analyzed the data, authored or reviewed drafts of the article, and approved the final draft.
- Renhe Zeng performed the experiments, authored or reviewed drafts of the article, and approved the final draft.
- Jianbin Ye conceived and designed the experiments, performed the experiments, analyzed the data, prepared figures and/or tables, and approved the final draft.

### Human Ethics

The following information was supplied relating to ethical approvals (i.e., approving body and any reference numbers):
The ethics committee of the Affiliated Hospital of Putian University (No:202183)

### Data Availability

The sequences are available at figshare: Ye, Jianbin (2023). Sequences-rawData-submitted.zip. figshare. Dataset. https://doi.org/10.6084/m9.figshare.22340578.v1
The sequences are available at NCBI: PRJNA928221.

### Supplemental Information

Supplemental information for this article can be found online at http://dx.doi.org/10.7717/peerj.15662#supplemental-information.

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
