# Peer review of "Alteration of the oral and gut microbiota in patients with Kawasaki disease"

_PeerJ, doi:10.7717/peerj.15662_

## Round 0.1 · original submission · Minor Revisions

Overall, the manuscript is well written and highlights the importance of gut microbiota in KD pathology and prognosis. However, there are few concerns that need to be addressed before the manuscript could be considered for publication. The authors are suggested to fix typo and grammatical errors throughout the text. Figures in the results section need to be improved in terms of clarity and labeling. There is an inconsistency found in the severity of KD in patients used for the study. Overall results need more justification due to the very low sample size. This should be included as limitations and discussed.

Reviewer 1 ·

Basic reporting

The authors have done a great job in writing this manuscript. Their efforts to provide in-depth details of microbiota from KD patient is invaluable. Their report is the first of its kind, giving new data on oral microbiota. Their selection of sample collection is unique. Their literature review is current and supplementary data is very clear and easy to understand. The writing is excellent. Their data and article will provide more insight into the scientific field.

Experimental design

No comment

Validity of the findings

No comment

Additional comments

However, they need to do minor editing in the manuscript,
1. If they can provide the limitations of their study that would be beneficial to understand.
2. Please, look at the spelling mistake in lines 119, 169, 174, 175, Figure 1 page 21, figure 2 page 23, figure 3 page 25, Figure 4 page 26, Figure 25 page 27 Where I believe that the word “Health” means “Healthy”. Kindly, please correct this in order to avoid more confusion while reading.

Reviewer 2 ·

Basic reporting

Please see the attached document

Experimental design

Please see the attached document

Validity of the findings

Please see the attached document

Annotated reviews are not available for download in order to protect the identity of reviewers who chose to remain anonymous.

·

Basic reporting

The manuscript articulates the importance of the gut and oral microbiota in patients with Kawasaki disease. The study is novel and will help in the differential diagnosis of KD. The authors have sufficient background and adequate literature support to compare study findings. Plus the results are relatable and provides light on the requirement to test oral and gut microbiota in patients in KD to understand the disease etiology and may be useful for prognosis.

Experimental design

The research question is well-defined and important to under the cause of rare diseases. Being a rare disease, the sample size of 5 although small to draw definitive conclusions is still valid to understand the microbiota distribution in comparison with controls. The investigation is sound and the techniques used are apt for evaluation. The methods have been clearly explained and detailed with sound statistics.

Validity of the findings

The results are novel and well articulated. The data supports the findings and study conclusions. The minor issue is the authors claim the KD patients were in the Acute phase of KD in lines 120/121. However, the authors in the discussion section claim KD patients are both Acute and non Acute in line 276. Not sure which is the correct categorization and the additional explanation on the phases of KD will be helpful for readers. Plus the age and how long the patients were diagnosed with KD will be helpful for understanding the KD pathogenesis. The conclusions are bold and with the sample size of 5 KD patients need to be used with caution. Requires some replication in other settings and population to validate the results.

Reviewer 4 ·

Basic reporting

No comments.

Experimental design

1. One weakness of the study is the sample size, which is only 5 patients and 5 healthy controls.
2. Previous study showed that the gut microbiota significantly differed between acute and non-acute stages in KD patients. Thus, as a follow up study, it is suggested to analyze the role of microbiota in the disease stages and severity level.
3. They could potentially analyze the level of inflammatory markers, e.g., IL-2, IL-4, IL-6, IL-10, TNF-α, INF-γ in the patients’ samples to see how the specific microorganism can influence the inflammation level in KD patients.

Validity of the findings

No comment.

Additional comments

Comments:
The article entitled “Alteration of the oral and gut microbiota in patients with Kawasaki disease” by Zeng et al, submitted to the journal PeerJ Life & Environment is very well written, informative, and well structured.
Strength:
They have studied the role of fecal and oral microbiota in Kawasaki disease (KD) patients and compared those with healthy controls. They took the opportunity of using high throughput sequencing technique to analyze the diversity and composition of microbiota and showed how microbiota diversity is associated with the disease.
Weakness:
1. One weakness of the study is the sample size, which is only 5 patients and 5 healthy controls.
2. Previous study showed that the gut microbiota significantly differed between acute and non-acute stages in KD patients. Thus, as a follow up study, it is suggested to analyze the role of microbiota in the disease stages and severity level.
3. They could potentially analyze the level of inflammatory markers, e.g., IL-2, IL-4, IL-6, IL-10, TNF-α, INF-γ in the patients’ samples to see how the specific microorganism can influence the inflammation level in KD patients.
After critically reviewing this article, it is strongly suggested to the authors to address the below:
1. Please provide the information for: age, sex, any history of pre-existing diseases including gastrointestinal diseases, whether they had prolonged fever during the time of inclusion in the study.
2. The intake of pro-biotics, antibiotics significantly influence the gut microbiota. In the sample collection section, they mentioned that no patients were taking antibiotics during the time of sample collection. But it is strongly recommended to provide information regarding the history of taking any antibiotics, probiotics withing 3 months of prior to the inclusion of the patients, controls in this study.
3. Please fix the minor typing mistakes: line 139 should not end with “by”, in lines 119, 174 and 175 the “health control” should be “healthy control”.

Annotated reviews are not available for download in order to protect the identity of reviewers who chose to remain anonymous.

---

## Round 0.2 · accepted · Accept

The authors have adequately addressed all the comments and the manuscript is ready for acceptance.

Reviewer 1 ·

Basic reporting

The paper has improved its quality by addressing the comments.

Experimental design

no comment

Validity of the findings

no comment

Additional comments

They have addressed all the comments that I asked them to address.

Reviewer 2 ·

Basic reporting

Authors have addressed all the concerns raised in the first review process. They have modified the text as well as the concerned figures and added extra information to clarify and support their findings. This article should be accepted with all the modifications.

Experimental design

No comments

Validity of the findings

No comments

Additional comments

No comments

·

Basic reporting

The manuscript is well-written and the study findings are novel. The study shows the importance of gut microbiota in Kawasaki disease pathogenesis. The results are well explained with sufficient discussion points comparing other reported studies.
After a thorough revision, I believe the manuscript is good for further process of publication. The authors have addressed all comments and revision requests made by the reviewers.

Experimental design

No comment

Validity of the findings

No comment

Reviewer 4 ·

Basic reporting

Nothing to report.

Experimental design

Nothing to suggest further.

Validity of the findings

No further comments.

Additional comments

Thank you addressing all the comments made by the reviewers. The revised manuscript is improved significantly and does not have any other issues to address.